# Automated Georectification, Mosaicking and 3D Point Cloud Generation Using UAV-Based Hyperspectral Imagery Observed by Line Scanner Imaging Sensors

Anthony Finn *[ID], Stefan Peters [ID], Pankaj Kumar and Jim O'Hehir

Science Technology, Engineering and Mathematics, University of South Australia, Mawson Lakes, Adelaide, SA 5095, Australia; stefan.peters@unisa.edu.au (S.P.); pankaj.kumar2@unisa.edu.au (P.K.); jim.ohehir@unisa.edu.au (J.O.)
* Correspondence: anthony.finn@unisa.edu.au

**Abstract:** Hyperspectral sensors mounted on unmanned aerial vehicles (UAV) offer the prospect of high-resolution multi-temporal spectral analysis for a range of remote-sensing applications. However, although accurate onboard navigation sensors track the moment-to-moment pose of the UAV in flight, geometric distortions are introduced into the scanned data sets. Consequently, considerable time-consuming (user/manual) post-processing rectification effort is generally required to retrieve geometrically accurate mosaics of the hyperspectral data cubes. Moreover, due to the line-scan nature of many hyperspectral sensors and their intrinsic inability to exploit structure from motion (SfM), only 2D mosaics are generally created. To address this, we propose a fast, automated and computationally robust georectification and mosaicking technique that generates 3D hyperspectral point clouds. The technique first morphologically and geometrically examines (and, if possible, repairs) poorly constructed individual hyperspectral cubes before aligning these cubes into swaths. The luminance of each individual cube is estimated and normalised, prior to being integrated into a swath of images. The hyperspectral swaths are co-registered to a targeted element of a luminance-normalised orthomosaic obtained using a standard red–green–blue (RGB) camera and SfM. To avoid computationally intensive image processing operations such as 2D convolutions, key elements of the orthomosaic are identified using pixel masks, pixel index manipulation and nearest neighbour searches. Maximally stable extremal regions (MSER) and speeded-up robust feature (SURF) extraction are then combined with maximum likelihood sample consensus (MLESAC) feature matching to generate the best geometric transformation model for each swath. This geometrically transforms and merges individual pushbroom scanlines into a single spatially continuous hyperspectral mosaic; and this georectified 2D hyperspectral mosaic is then converted into a 3D hyperspectral point cloud by aligning the hyperspectral mosaic with the RGB point cloud used to create the orthomosaic obtained using SfM. A high spatial accuracy is demonstrated. Hyperspectral mosaics with a 5 cm spatial resolution were mosaicked with root mean square positional accuracies of 0.42 m. The technique was tested on five scenes comprising two types of landscape. The entire process, which is coded in MATLAB, takes around twenty minutes to process data sets covering around 30 Ha at a 5 cm resolution on a laptop with 32 GB RAM and an Intel® Core i7-8850H CPU running at 2.60 GHz.

**Keywords:** hyperspectral imaging; mosaicking; georectification; UAV; pushbroom





## 1. Introduction

Hyperspectral techniques were originally used in Earth observation applications and represented a revolution in satellite-based remote sensing. Within the past decade, miniaturisation has allowed such sensors to be mounted on unmanned aerial vehicles (UAVs). This offers opportunities for examining the spectral behaviour, and thus surface chemistry, of surveyed objects at a high spectral (<5 nm) and spatial resolution (<10 cm)

and low operational cost. The range of applications includes agriculture [1–3], forestry [4] and mining [5]. For instance, [6,7] investigated the early detection of plant diseases and the seasonal trends of narrow-band physiological and structural vegetation indices, and [8–10] collected high resolution hypercubes to map the health and status of vegetation.

UAV hyperspectral sensing has matured rapidly in recent years [11,12]. However, its routine application has been traditionally constrained by a lack of automation for streamlining the image analysis [13]. That is, until recently a great deal of effort has been required to manually co-register imagery and ground control points (GCPs). To realise their full potential, UAV-based hyperspectral sensing systems need to provide radiometrically and geometrically accurate observations that can be post-processed quickly and with confidence. However, despite significant progress in fusing the hyperspectral data acquired by UAVs [8,11,14–16], there appears to be no method that automatically, rapidly and robustly georectifies and mosaics hyperspectral data cubes and then converts the result into 3D hyperspectral point clouds. Moreover, in an ideal world, such a program would operate in *batch* mode on a 'standard' laptop, i.e., the software would process data with minimal human intervention on modest computational facilities. In terms of benefits, as well as adding efficiency to the data processing of scenes observed using pushbroom hyperspectral sensors, the 3D point cloud adds geometric features to any analysis. That is, it helps in the examination of hyperspectral data distribution relative to reconstructed surfaces, and thus informs the approach to non-trivial tasks such as modelling complex surface reflectance characteristics, including specular reflections, inter-reflections, transparencies and sub-surface scattering under real-world conditions.

In this paper, we propose a processing pipeline for swiftly and automatically generating, fusing, georectifying and rendering dense hyperspectral point clouds. The proposed methodology aims to enable hyperspectral analysis in 3D space. The main contributions of the paper may be summarized as: (i) subtle refinement of a computationally robust georectification and mosaicking technique proposed by Angel et al. [8] for geometrically transforming and merging hyperspectral scanlines into a single spatially continuous mosaic, and (ii) expansion of that approach by converting these accurate georectified 2D hyperspectral mosaics into 3D hyperspectral point clouds. Crucially, the technique described here substantially reduces both the raw pre-processing requirements and computational load of the workflows required by previous researchers, thereby increasing the method's degree of automation and hence operational utility.

To reduce the pre-processing requirements, the proposed method morphologically and geometrically assesses (and, if possible, repairs) hyperspectral data cubes before aligning them into linear swaths. To reduce the processing load in the substantive workflow, geometrically based pixel index manipulation is used rather than 2D convolution. Luminance across the RGB and hyperspectral data sets is also normalised to reduce the effects of differing illumination conditions and thus improve the georectification performance.

The paper is organised as follows: prior work in the field is summarised in Section 2, which allows the reader to understand the state of the art; in Section 3, the proposed processing pipeline for generating hyperspectral point clouds is explained; experimental results are described in Section 4; and the conclusions drawn by the development of this work are presented in Section 5.

## 2. Related Work

Digital cameras find many applications in remote sensing. However, the spectral information contained in their imagery is typically limited to only three bands (red, green and blue) for colour images. Multispectral cameras partially overcome this limitation, but these sensors are also restricted in the number of frequency bands they make available per frame. In contrast, hyperspectral cameras provide potentially hundreds of spectral bands per pixel. The use of these cameras has therefore significantly increased in recent years. This is mainly due to the availability of more cost-effective sensors and versatile platforms [11]. Systems are also improving in terms of accuracy, efficiency, signal-to-noise ratio and ease

of use; and of these systems, pushbroom sensors are the most widely available. In part, this is because pushbroom sensors do not require a complex mechanical scanning mechanism. Those instruments that do employ scanning mechanisms are generally known as whiskbroom sensors [12,17] and suffer from increased weight, volume and cost.

Pushbroom sensors [18–24] offer a high spectral and spatial resolution by sampling individual spectral lines. However, such instruments are effectively line scanners, with their spatial accuracy highly dependent on flying conditions and the stability provided by any host platform and gimbal. The resulting errors are thus constrained by the accuracy of the navigation sensors, which are required to compensate for image distortion that often results from the geometric noise induced by a UAV's motion [4,11,25].

In an attempt to address this, several researchers have investigated methods that make use of multiple pushbroom sensors [26–28]. Similarly, lower spectral and spatial resolution multi-spectral cameras also offer a valuable alternative [11,29–33]. These cameras collect band-sequential spectra in two spatial dimensions or by integrating multiple synchronised cameras. In both cases, the mosaicking process is able to make use of photogrammetric techniques such as structure from motion (SfM) [34–37], which provides spatial accuracy in their image products. This makes them well-suited to campaigns in real-world environments. As a result, the routine use of hyperspectral pushbroom systems remains challenging, in part due to the lack of automation and processing options available [38].

Notwithstanding, several researchers have developed techniques for classifying materials and artefacts [39–42] using hyperspectral imagery. However, despite the radiometric accuracy of these techniques, the resulting images typically display considerable geometric deformation, making their combination with other data sources difficult.

High-precision stitching with small overlaps is important for many applications; and accurately overlapping swaths typically requires a large number of GCPs to avoid further distortion in the final mosaic [11]. Some researchers have attempted ortho-rectification based on Global Navigation Satellite System/Inertial Navigation System (GNSS/IMU) parameters, a digital elevation model (DEM) and the PARGE software [43]. Others have used a dense network of GCPs in addition to PARGE [22,44]. The studies report that a 5 cm root-mean-square error (RMSE) was achieved using hyperspectral imagery of a 2–4 cm ground resolution distance (GRD). However, the field deployment of GCPs is time-consuming and the PARGE software costs up to AUD 8500 for a node-locked licence.

Traditionally, additional time-consuming effort is also required during the post-processing stages to overcome the geometric distortions introduced though any UAV movement during flight. To address this problem Angel et al. [8] developed a computationally robust automated georectification and mosaicking methodology that operates in a parallel computing environment running on a high-end Intel Xeon E5-2680 v2 processor (20 cores running at 2.8 GHz with 200 GB of RAM). They evaluated their results against several high-spatial-resolution (mm to cm) data sets collected using a pushbroom hyperspectral sensor and an associated RGB frame-based camera. The technique estimates the luminance of the hyperspectral swaths and co-registers these against an RGB-based orthomosaic using the speeded-up robust features (SURF) algorithm to identify common features between each swath (derived during a pre-processing stage) and the RGB-based orthomosaic. The maximum likelihood estimator sample consensus (MLESAC) approach then finds the best geometric transformation model to the retrieved feature matches. Individual scanlines are thus geometrically transformed and merged into a single spatially continuous mosaic using only a small number of GCPs. The technique offers processing speed improvements for a co-registration of 85% relative to the traditional manual/user-generated procedures.

Although the results provided by this method are excellent, and it is a great advance over previous approaches, it should be noted that considerable computational power and several hours were needed to process the data. As a result, the significant resources demanded may preclude its broader use. Moreover, the raw hyperspectral swaths used in the feature matching steps were generated during an initial pre-processing stage rather than during the (timed) MATLAB workflow described in the paper. The researchers also

concluded that different illumination conditions and the use of different sensors (RGB vs. hyperspectral), resulting in distinct geometries, influence the number of features detected in each data set, and thus the georectification performance.

In regard to the creation of 3D point clouds, light detection and ranging (LiDAR) has been integrated with the hyperspectral sensors [45,46]. Such combinations allow both data sets to be synchronised, mitigating UAV platform positioning errors with respect to sensor resolution and any platform instability in flight. They also allow the hyperspectral characterisation of solid objects. Despite promising results, LiDAR sensors, even those considered low-cost, are expensive and require UAVs with a higher payload capacity. Consequently, such integrations are not yet readily accessible to everyone.

As a result, a number of researchers have developed other approaches to obtain models with information fed from different sensors. For instance, Ref. [47] developed a technique that applies SfM to images from different wavelengths and a 3D registration method to combine band-level models into a single 3D model. Similarly, ref. [48] describes an out-of-core method that enables the on-site processing and visualization of data by using a collaborative 3D virtual environment. Other works have merged hyperspectral data with 3D models generated from standard RGB cameras or laser scanning to model geological structures, map underwater environments and carry out plant phenotyping [49–51].

The processes needed to deal with the large quantities of 3D data and hyperspectral images are usually computationally demanding, and [52] describes an approach based on the efficient generation of hyperspectral point clouds with a low-memory footprint. The procedure is run on a GPU using OpenGL's compute shaders to build and render the 3D hyperspectral point cloud.

In view of the above, and in line with the goals espoused by Angel et al., the aim of this research was "to develop an automated workflow capable of producing accurate georectified UAV-based hyperspectral mosaics collected by pushbroom sensors", and to extend the workflow of [8] by creating 3D hyperspectral point clouds. In addition to adding geometric features to any image analysis, the generation of a 3D model that is enriched with hyperspectral data helps in the study of the distribution of spectral information relative to surface orientations under real-world conditions.

Thus, the aim of this research is to minimise human intervention during the geoprocessing stage, avoid reliance on GCP and fully automate an efficient "batch processing" co-registration, mosaicking and point cloud generation strategy. The goal was also to reduce the computational requirements of any technique such that the algorithms could run rapidly on readily available computing equipment.

The methodology proposed here improves a technique that provides a solution for one of the most time-consuming post-processing stages of UAV-based hyperspectral remote sensing using pushbroom sensors. It implements a simplified co-registration and mosaicking strategy and generates positionally accurate 3D hyperspectral point clouds; and it does so in a computationally efficient manner.

### 3. Materials

*Study Areas:* Data were collected from a number of sites around Adelaide, South Australia. The first data set supports a study undertaken to examine the detection and location of young trees [53]. The data were observed on consecutive days (20/21 October 2019) over two sites, known as Goat Farm A and B. Goat Farm is located within the Mount Crawford Forest area, South Australia (34.7137°S 139.0238E, 425 m elevation). An orthomosaic image of the site at Goat Farm A is shown in Figure 1.

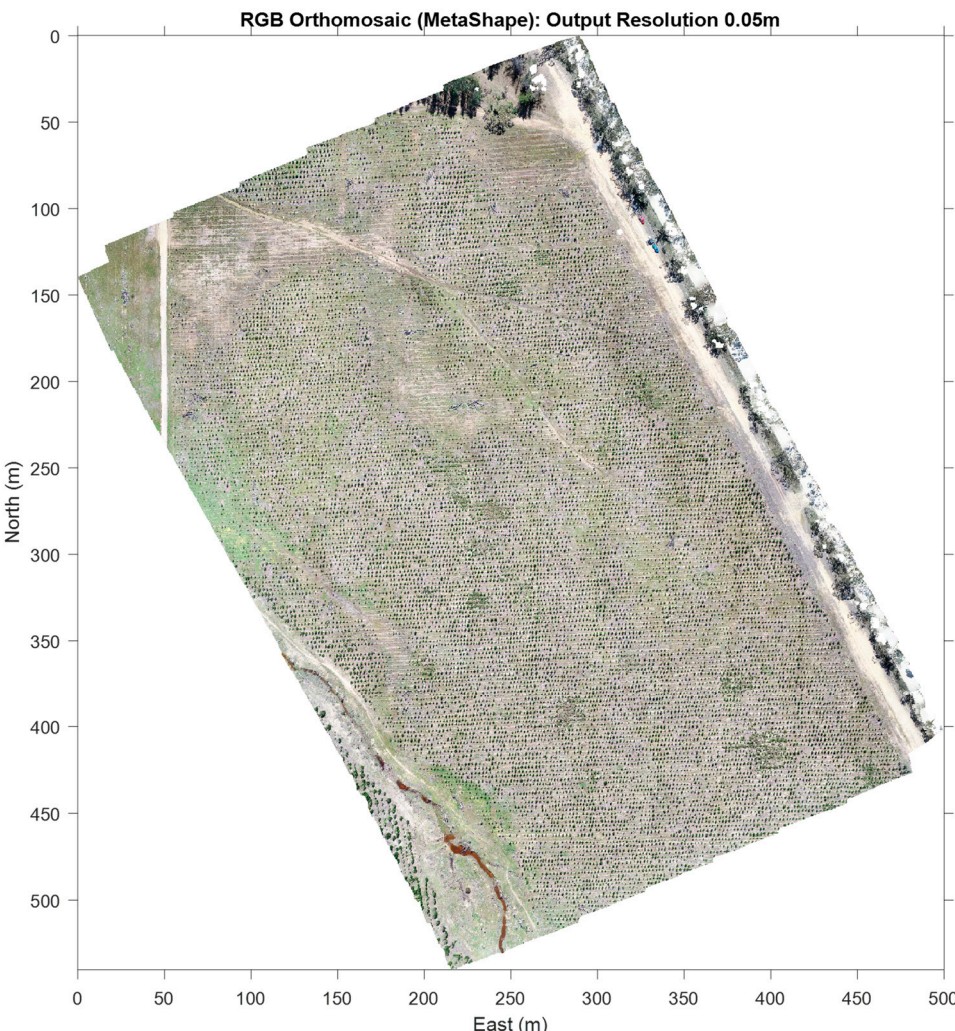

**Figure 1.** Orthomosaic of the site Goat Farm B.

Both A and B comprise *Pinus Radiata* seedlings. Site A contained trees aged about 14 months old (maximum tree heights/crown diameters about 40 cm/30 cm, respectively). Site B contained trees of about 3 years old (maximum tree heights/crown diameters of 2.5 m/1.5 m, respectively). Site A occupied about 13 Ha, Site B around 15 Ha. Both had stocking levels of around 1400 trees/hectare and comprised a similar geomorphology: some weed contamination, coarse wooden debris and a sharply positive and negative local topography. The UAV flights reported here took place between 10 a.m. and 2 p.m. local time. There was no precipitation.

The second data set, taken from what are referred to as the PEGS sites, was observed between 12 December 2019 and 29 January 2020. These data were observed in support of a study into the complex relationship between local ground temperature, flora density/height/prevalence and distance from the edge of the scrub (unpublished). This data set comprises three sites, referred to as Kaiserstuhl, Bonython and Hale, which had a similar geomorphology: native eucalypt trees and low shrubs covering undulating hills. There were roads and other manmade objects (tracks and buildings) nearby. An orthomosaic image of the Hale site is shown in Figure 2. As with the Goat Farm data, the UAV flights took place between 10 a.m. and 2 p.m. local time. Once again, there was no precipitation. Each site covered between 10 and 15 Ha.

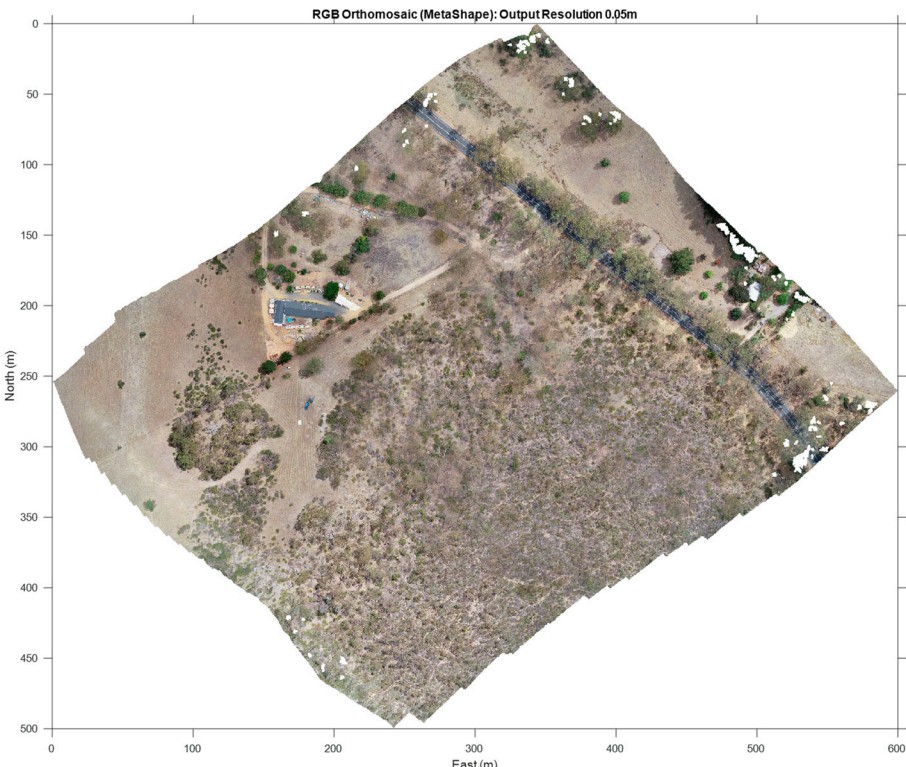

**Figure 2.** Orthomosaic of the PEGS site Hale.

***Hyperspectral Data Acquisition:*** All hyperspectral imagery in this study was collected using a DJI Matrice M600 hexacopter (www.dji.com), coupled with a Ronin-MX gimbal to reduce the effect of UAV flight dynamics. The Matrice 600 carried a Resonon Pika-L pushbroom camera (www.resonon.com/pika-l), with a 17 mm lens and a horizontal field of view (FOV) of 17.6°. The Pika-L gathers radiometric data at a sampling rate of 249 frames (lines) per second across up to 281 bands in the near infrared (NIR) band, which extends from 400 to 1000 nm. It has 900 spatial pixels per line.

The hyperspectral sensor draws navigation data from its own GNSS-IMU sensor to monitor the pose (position and orientation) of the camera during flight. All flight paths followed standard "lawnmower" patterns over the sites for which the intended side overlap between swaths was <2%. The reasons for using such narrow overlaps are outside the scope of this study. However, this choice may have been serendipitous as the resultant mosaicking indicates that operational gains may potentially be on offer. The approach appears to need only small amounts of overlap between swaths (just enough to avoid gaps created by unexpected aircraft motion), which would mean reduced flight times and/or a greater area coverage per flight. However, this does mean that aircraft motion occasionally caused gaps between contiguous swaths in this data set.

The forward speed of the UAV during flight was around 5 m/s and the gimbal orientation was set to follow the forward direction of travel at all times. The UAV travelled at an altitude of 60 m, which provided an along- and across-track GRD of about 2 cm. This roughly matched the across-track GRD of the sensor.

***RGB Image Acquisition:*** All RGB images used in this study were acquired using a DJI Phantom IV quadcopter (www.dji.com). The altitude of the hyperspectral aircraft was 60 m at both Goat Farm and the PEGS sites. All flight paths followed standard 'lawnmower' patterns over the sites, with images obtained from the gimballed camera capable of image capture in the red (615–695 nm), green (525–570 nm) and blue (416–476 nm) bands. The Phantom IV carried a 1″ CMOS, 20 Mpix, 2.4 μ, 77° FOV camera, providing a native image GRD of 1.5 cm for both the Goat Farm and PEGS sites. The orientation and forward velocity of the UAV was fixed and constant, and at least 10 m/s on all flights. Images were taken

at a frame rate that allowed an overlap of about 80% in the along-track direction and 60% in the across-track direction. The accuracy of the imagery observed by the Phantom IV relied upon the stand-alone coarse acquisition global navigation satellite system (GNSS), i.e., around ±5 m. GCPs were deployed but were not used in this study.

## 4. Methods

The automated georectification, mosaicking and 3D DPC generation workflow described in this paper is shown in Figure 3. It was fully coded in MATLAB 2023a and ran on a Dell laptop with 32 GB RAM and Intel(R) Core i7-8850H CPU running at 2.60 GHz. The following sections describe the workflow.

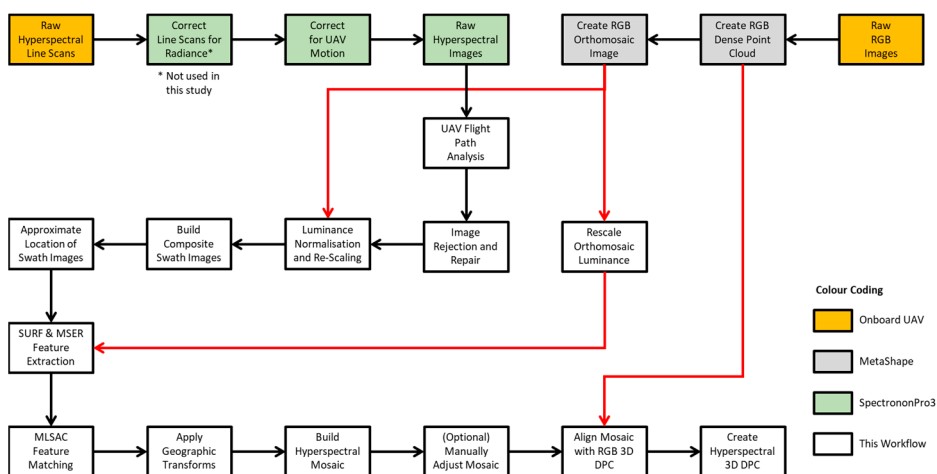

**Figure 3.** Workflow of methodology employed in this study. Colour code indicates the software packages used to create output (red lines indicate output is derived from the MetaShape). All software uses batch mode processing to minimise user effort.

*Generation of RGB Orthomosaic and Dense Point Cloud:* The sequence of Phantom IV images observed at each site were converted into a 3D dense point cloud (DPC) and an orthomosaic image using Agisoft's MetaShape v1.8.3 software (www.agisoft.com). MetaShape uses SfM, which consists of two main steps: camera motion estimation and reconstruction of the DPC. Initially, a sparse set of points are matched across the acquired 2D image stack to determine correspondences. Such features are typically extracted using algorithms like SIFT (scale invariant feature transformation) [54] and SURF (speeded-up robust features) [55]. The end result is a collection of $n$ points (where $n$ is typically of the order of $10^8$), represented by the spatio-spectral function, $g(x_i, y_i, z_i, R_i, G_i, B_i)$, for $i = 1, \ldots, n$.

MetaShape offers varying levels of point cloud (and hence orthomosaic) resolution: ultra-high, high, medium and low. The point clouds used in this study were generated using "high" levels of resolution, which typically represent a text file size of around 8–10 GB and a DPC density of about 3–4 cm (about $10^8$ points). The TIF images, created as orthomosaics from the DPC with 2 cm GRD, typically have file sizes of about 2–3 GB. MetaShape uses a bundle adjustment to initially align the images (40,000 key points, 4000 tie points).

Although the GRD of the orthomosaic RGB image and hyperspectral data were around 1–2 cm, both were resampled to a common resolution of 5 cm. Bi-linear interpolation was used, with the value of the output pixel estimated from an average of its four closest neighbours. This ensured the processing would run on a 32 GB RAM computer. At least 64 GB RAM was required to run the data for the 15 Ha plots at a higher 2.5 cm GRD. In comparison, ref. [8] made use of ultra-high levels of resolution, and while the tomato and date palm plots mosaicked were smaller, the observed resolutions were higher (the UAV flights were at a lower altitude in [8]). Overall, however, the number of pixels generated per georectified mosaic in both studies were of a similar order, i.e., around $10^8$ pixels/mosaic

and $10^6$ pixels/hyperspectral image. The number of UAV flight lines were also similar (11–15 flight lines per site in this study).

*Raw Hyperspectral Data Processing:* Pushbroom sensors deliver raw pixel data in the form of rows and columns. In other words, the raw data have no pre-associated geographic frame of reference. Such images must first be corrected using a process known as georectification to remove the geometric and location distortions. During this process, pixels are first transformed to a common plane to correct for geometric distortions. They are then georeferenced, i.e., real-world geodetic coordinates are assigned to each pixel in the image.

This initial correction was performed using SpectrononPro3 v3.4.4 software (www.resonon.com), which can be operated in batch mode such that it outputs individual TIF images and/or data cubes, both corrected for distortion induced by the camera's 3D motion. Unfortunately, GNSS-IMU sensor errors lead to geometric errors in the pre-processed output cubes/images, which is why additional processing is needed. Errors also arise from other sources, including mechanical and optical imperfections in the sensor, atmospheric effects and imperfections in the digital terrain (or flat earth) model. Importantly, SpectrononPro3 includes the WS84 coordinates of each corner of the image/cube (henceforth referred to only as cubes). As with the distortion corrections, however, these coordinates are contaminated with errors.

TIF images are generated from the corrected cube output comprising the three-colour RGB combination of the spectral bands (observed by the Pika-L sensor) closest to 670 nm (red), 540 nm (green) and 480 nm (blue). Although radiance and reflectance corrections can be applied, in this study they were not. As per [8], these individual images can then be mosaicked into swaths and processed as per the rest of this workflow, skipping the next two steps. However, in order to accord with [8]'s workflow, additional user effort is required to break contiguous raw data sequences—sets of consecutive cubes/files—into file sets commensurate with each swath; and even more effort is needed if the pitch, roll, yaw and timing offsets need to be optimised. To increase the degree of automation, we therefore introduced two simple steps: flight path analysis and image rejection and repair.

*Flight Path Analysis:* The flight path of the Matrice 600 UAV is analysed, and the corners of the lawnmower path automatically extracted. As the overall flight path of the UAV from take-off to landing can involve a considerable number of manoeuvres beyond those involved in the standard lawnmower pattern (e.g., to/from the lawnmower start/end points, rotations to orient the UAV along the line of flight, etc.), and there is a need for the approach to be robust to real-world data sets, this method is worthy of brief description.

First, the direction of travel, $D_k$, at each epoch, $k = 1 \ldots n$, is computed over the entirety of the flight; $k = 1$ being the epoch at which the first image is observed by the UAV and $k = n$ the last. A histogram of $D_k$ is formed and the bin with the most samples, $D_{Max}$, is deemed the primary direction of travel. The primary direction is referred to as $\Delta = 1$, and direction numbers, $\Delta_k = 1 \ldots 4$, are assigned to each epoch for which $D_k$ satisfies the inequality, $|D_{Max} + j.90 - D_k| < D_T$, where $D_T$ is a threshold (typically 10°) and $j = 0 - 3$ used to compute the four primary directions/quadrants of travel for a lawnmower pattern. $\Delta_k = 0$ is used to represent epochs during which the UAV is not travelling in any of the four primary directions. Thus, each epoch is assigned a parameter, $\Delta_k = 0 \ldots 4$, to indicate whether it is travelling (say) north, south, east or west, or none of these directions.

A set of change points, $C_i$, $i = 1 \ldots m$, are then computed by finding the epochs for which there is a change in the direction number, $\Delta_k$. Change points too close together, geographically or temporally, are excluded, as are those that form too straight a path to both their neighbours at epochs $k - 1$ and $k + 1$. Path lengths between the remaining change points (ostensibly corners of the lawnmower) are then computed and histograms are again used to find the most likely path lengths, which are assigned *long* or *short* based on their relative sizes. Simple checks are then applied to ensure the paths form a long–short–long sequence, start and end on a long path length, etc. This ensures any flight manoeuvres that precede the sequence of swaths used to overfly the site are not included in the processing (unless desired).

Using the centroids of the GNSS coordinates (Figure 4), $\left[x_{min}^{hyp_i}, y_{min}^{hyp_i}\right]$, $\left[x_{min}^{hyp_i}, y_{max}^{hyp_i}\right]$, $\left[x_{max}^{hyp_i}, y_{max}^{hyp_i}\right]$, $\left[x_{max}^{hyp_i}, y_{min}^{hyp_i}\right]$, which represent the corners of each hyperspectral image/cube, $i$, all of the cubes/images that align with each leg (long and short) of the UAV flight are then automatically grouped together into $n$ potential swaths. That is, images are combined and processed together in $n$ groups, $H_{j=1...m_i}^{i=1...n}$, where $m_i$ is the number of images in the $i^{th}$ swath (noting, despite the lawnmower flight path, due to the asymmetry of images during turns, each swath does not necessarily contain an identical number of images).

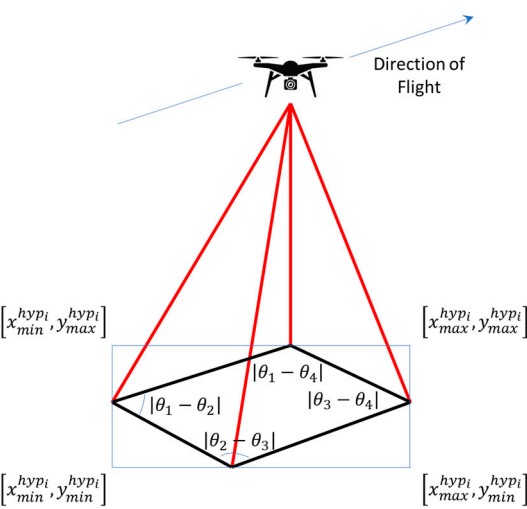

**Figure 4.** Graphical depiction of the hyperspectral image/cube, $i$. Images are combined and processed together into $n$ groups, $H_{j=1...m_i}^{i=1...n}$, where $m_i$ is the number of images in the $i^{th}$ swath and a swath is a linear combination of contiguous images.

The path is then identified from the image sequence using the assumption that each image obtained is file stamped by the hyperspectral sensor in monotonic order. That is, because the UAV often meanders before or after flying its designated (lawnmower) path or starts flying its intended path only to be tasked to re-start part-way through (so images observed during irregular flight phases are associated incorrectly (out of sequence) with the lawnmower pattern phase), the sum of the swaths, $\sum H_{j=1...m_i}^{i=1...n}$, is minimised. For speed, this is achieved using the number sequence of the image files.

*Image Rejection and Repair:* Generally, if a UAV is travelling at a constant forward velocity and environmental factors such as turbulence have little impact, the data cubes present as roughly rectangular entities. However, if the UAV rotates quickly at the end or start of a swath or environmental factors cause the UAV to accelerate for some other reason and the hyperspectral camera's navigation sensors do not update quickly enough, the raw georectification processing can introduce distortions into the imagery. These distortions typically manifest as non-rectangularly shaped images (bulges or extensions on one or more of the image sides) and/or gaps or indentations in the continuity of the imagery, usually on the sides of the images.

*Repair:* Before image registration commences, each cube is morphologically and geometrically checked for integrity. First, a binary mask of the RGB image is created and the properties of all *eight-connected* structures in the image computed. An eight-connected structure is an object whose pixels are all connected to one another such that each pixel is surrounded by 8 others. If the image contains more than one structure (i.e., the image has been broken into multiple components) or the number of pixels in the binary mask falls below 99.9% of what should occupy a convex polygon described by the mask's boundary, an attempt to repair the image is made: bilinear interpolation is applied to "fill" the empty pixels before the morphological and geometric tests are applied.

*Rejection:* A *closing* operation is then applied to the binary mask, which removes any small gaps in the data (closing involves *dilation* followed by *erosion*, where dilation enlarges or thickens the structure foreground object and erosion thins it). Finally, the perimeter and area of each contiguous region of the image, $P_{Morph}$ and $A_{Morph}$, respectively, are computed. The boundary points of the largest structure and the four extrema of this boundary are identified, and a polygon is fitted to these.

The perimeter and area of the polygon, $P_{Poly}$ and $A_{Poly}$, respectively, are then also computed. Any images with ratios of $\left|1 - \frac{P_{Morph}}{P_{Poly}}\right| > 1\%$ or $\left|1 - \frac{A_{Morph}}{A_{Poly}}\right| > 5\%$ are considered insufficiently rectangular to be usefully integrated into a swath composite. Before they are rejected, however, another geometric test is applied. The angle, $\theta_j$ (where $j = 1\ldots4$), that describes the slope of each side of the polygon is computed as modulo $90°$, and the values $|\theta_1 - \theta_2|, |\theta_1 - \theta_3|, \ldots, |\theta_3 - \theta_4|$ are computed (Figure 4). If 3 or more such values fall below a threshold, $\theta_T$ (typically $10°$), the polygon is considered to contain two parallel sides and one orthogonal one and is thus retained. Any sequence of images containing only one image is also rejected.

***Luminance Normalisation:*** The luminance of each pixel, $L_{ortho_i}$ and $L_{hyp_i}$, respectively, is retrieved from both the coloured RGB orthomosaic and hyperspectral images using the standard formula defined by the National Television System Committee (NTSC) [56]. This retains the luminance but discards the hue and saturation information in the images. Unlike [8], however, the luminance is first modified using $L_{hyp} = L_{hyp_i}\hat{L}_{ortho}/\hat{L}_{hyp}$, where $\hat{L}_{hyp, ortho} = \frac{1}{n}\sum_{i=1}^{n} L_{hyp_i,ortho_i}$, for $L_{hyp_i,ortho_i} \neq 0$, i.e., $\hat{L}_{hyp}$ and $\hat{L}_{ortho}$ are the mean values of luminance of the non-zero pixels in the orthomosaic and hyperspectral images.

The white balance of the pixels is also modified in a similar way. That is, each RGB channel is corrected individually such that the RGB value of each pixel, $L_{R,G,B} = L_{R,G,B}\hat{L}_{hyp}/\hat{L}_{R,G,B}$, where $\hat{L}_{R,G,B}$ represents the mean values of red, green or blue non-zero pixels, respectively, in the hyperspectral image. The modified values of $L_{R,G,B}$ are then concatenated into a three-colour RGB image. As the RGB values of each hyperspectral image output by SpectrononPro3 are maximally scaled within the range 0–255, the luminance of the orthomosaic pixels, $L_{ortho}$, are similarly rescaled such that their minimum and maximum values fall between 0 and 255. This accounts for (partial) variation in lighting conditions that can occur between the RGB and hyperspectral sensor flights.

***Creation of Composite Swath Images:*** In [8], the hyperspectral swaths are created during the pre-processing stage of the raw data. This can require considerable user effort as the start and end images of each swath must be identified and any corrupted hyperspectral imagery excluded from the workflow. The individual swaths must then be mosaicked and named (often sequentially) for subsequent processing.

To reduce user effort and avoid computationally intensive convolution operations that approximately locate the swaths within the orthomosaic, we create a 3D matrix, of dimensions $[p, q, r]$, where $p = \left(x_{max}^{ortho} - x_{min}^{ortho}\right)/GRD$, $q = \left(y_{max}^{ortho} - y_{min}^{ortho}\right)/GRD$ and $r = 3$; and $x_{max}^{ortho}, x_{min}^{ortho}, y_{max}^{ortho}, y_{min}^{ortho}$ represent the extrema of the orthomosaic image and *GRD* the desired ground resolution distance at which the image matching is to be conducted (0.05 m in this study). The third dimension of the matrix, $r$, has one channel reserved for each band of the RGB triplicate.

Commencing at the first image of the first swath, $H_1^1$, each of the $m_1$ (Figure 5) hyperspectral images are processed in turn. First, the extrema, $x_{max}^{hyp_1}, x_{min}^{hyp_1}, y_{max}^{hyp_1}, y_{min}^{hyp_1}$, are computed from $\left[x_{c_{1\ldots4}}^{hyp_1}, y_{c_{1\ldots4}}^{hyp_1}\right]$, the coordinates of the corners of the hyperspectral image. Then, the image is resampled into a matrix of dimensions, $[u, v, w]$, where $u = x_{max}^{hyp_1} - x_{min}^{hyp_1}/GRD$, $v = y_{max}^{hyp_1} - y_{min}^{hyp_1}/GRD$ and $w = 3$. Next, this resampled matrix is allocated to a sub-set of pixels, $[a : b, c : d]$, where $a = x_{min}^{hyp_1}/p$, $b = x_{max}^{hyp_1}/p$, $c = y_{min}^{hyp_1}/q$ and $d = y_{max}^{hyp_1}/q$, within an (initially blank) matrix of dimensions $[p, q, r]$. Although this is an inaccurate placement, not least because the corners of the hyperspectral images $H_j^i$

do not form a rectangle, this is performed because it is computationally much faster to allocate a rectangular $p \times q \times r$ sub-block of pixels in memory than to accurately assign, pixel-by-pixel, a slightly tilted, trapezoidal image to the correct elements within the relevant matrix needed to form part of the composite image representing each swath.

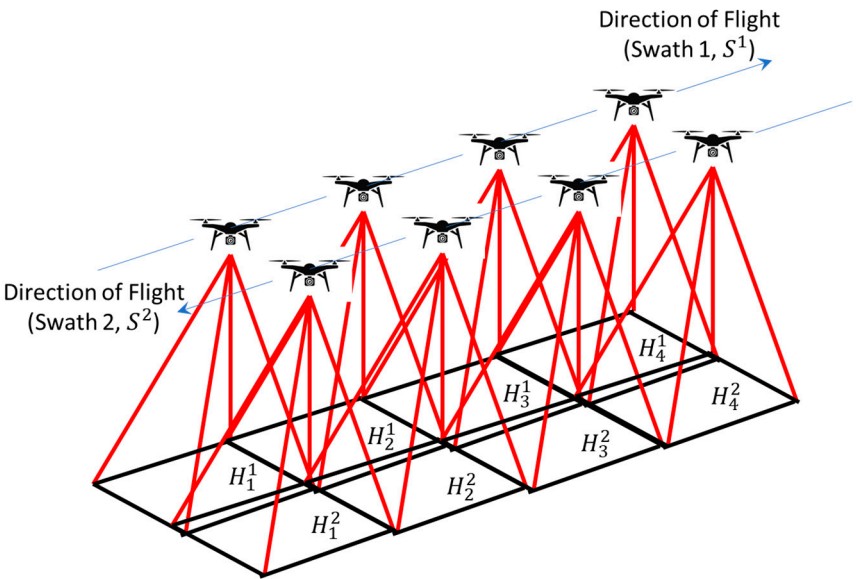

**Figure 5.** Graphical depiction of swaths, comprising hypercubes/imagery, $H_1^1 \ldots, H_{14}^2$.

However, it also means the image formed by the $p \times q \times r$ sub-block of pixels, $H_j^i$, is slightly distorted relative to its true orientation. $H_j^i$ is therefore transformed, using an affine transformation, from its position described by the coordinates, $\left[ x_{min}^{hyp_1}, y_{min}^{hyp_1} \right]$, $\left[ x_{min}^{hyp_1}, y_{max}^{hyp_1} \right]$, $\left[ x_{max}^{hyp_1}, y_{max}^{hyp_1} \right]$, $\left[ x_{max}^{hyp_1}, y_{min}^{hyp_1} \right]$, to its correct orientation, $G_j^i$, defined by the corner coordinates of the image, $\left[ x_{c_{1...4}}^{hyp_1}, y_{c_{1...4}}^{hyp_1} \right]$. Executing a transform of $H_j^i$ at this stage, and then combining the transformed images, $G_j^i$, into swaths rather than attempting a block transform $S^i$ to its transformed version, $T^i$, (to account for both the georectification errors in the swath and the orientation errors in the individual images), delivers an improved performance. This is because the global transformation applied to $S^i$ is typically unable to account for the individual (small) adjustments in orientation required by each of the individual images, $H^i$.

For hyperspectral images $H_{j=2\ldots m_i}^{i=1\ldots n}$, it is assumed that the swaths were formed as a result of a (more-or-less) constant velocity forward flight by the UAV. In other words, it is assumed that each image is a contiguous extension of the previous one, and that swaths can be reconstituted as a single continuous image from their component images. Consequently, after the creation of $G_{j=2\ldots m}^{i=1\ldots n}$, the closest corners $\left[ x_{c_{1...4}}^{hyp_j}, y_{c_{1...4}}^{hyp_j} \right]$ to the previous one $\left[ x_{c_{1...4}}^{hyp_1}, y_{c_{1...4}}^{hyp_1} \right]$ are identified and differenced, and the offsets, $\Delta x_i^{1,2}$, $\Delta y_i^{1,2}$, added to corners of the relevant coordinates of subsequent images. This enables the small errors in the GNSS coordinates recorded by the pushbroom sensor to be aligned such that the swaths form a single continuous image.

*Approximate Location of Swath:* As $S^i$ is now located reasonably accurately within the pixel index framework of the orthomosaic, extensive computational effort is not required to find matching features between the two data sets. A logical mask, $M^i$, of $S^i$ is created and a large dilation operation conducted. A mask size of 5–10 m (converted to pixels by dividing by GRD) typically proved sufficient. The resulting (larger) mask, $M^{*i}$, is now larger than $M^i$ and $S^i$, but still covers the correct location, whilst also accounting for GNSS-IMU navigation sensor errors between the (potentially different) UAVs carrying the

RGB and hyperspectral pushbroom sensors. The mask $M^{*i}$ is then applied to the whole orthomosaic, such that the resulting partial orthomosaic, $O^i$, and hyperspectral swath, $S^i$, can be accurately georectified in a manner that maximises computational efficiency. In other words, the prospect of matching features within $S^i$ with those outside its equivalent section of the orthomosaic is minimised as the two images contain (almost) all of the same features (edges, corners, blobs, ridges, etc.).

***Extraction of Matching MSER and SURF Points:*** Maximally stable extremal regions (MSER) [57] and speeded-up robust feature (SURF) [58] extraction techniques were jointly employed to detect key features within the orthomosaic and hyperspectral swaths. The SURF implementation is that implemented in MATLAB and thus similar to that described in Section 3.4 of [8], and readers are referred to that paper for a description of the mathematics involved and the rationale behind the technique's use. Although SURF is a scale-invariant feature detector, it is known to offer a much better performance when the features are of comparable scale and the luminance values similar [59].

As SURF detects features based on textural analysis, it will identify very few features from an image that contains flat homogenous terrain covered by little or no vegetation. In such a case, MSER feature extraction function is added as MSER incrementally steps through the luminance range of the images to detect stable intensity regions. MSER therefore complements the SURF approach by performing well over homogenous areas of geomorphology [60]. This increases the number of true feature-matches likely to be identified between scenes in the event they were captured under slightly different lighting conditions.

The MSER implementation is that provided by MATLAB and the key parameters used by MSER and SURF include: the "strongest feature threshold", "number of octaves" and "feature size". The values of these parameters used in this study were 100, 1 and 64, respectively. Their values were derived empirically, and readers are referred to [8,58–60] for more details.

Prior to any feature detection, however, a 0.5-pixel 2D Gaussian smoothing kernel is applied to the input images as this slightly improves the inlier feature matching performance of MLESAC.

***Selection of Matching Points by MLESAC:*** An MLESAC algorithm [61] was employed to ensure outlier-corrupted feature matches do not contaminate the derivation of the best transformation model. As per the previous section, readers are once again referred to [8] (Section 3.5) for a description of the mathematics and rationale behind MLESAC and its place in the workflow. The values of key parameters (see [8,61] for more details) such as "search radius" (search radius associated with identified centre points), "matching threshold" (threshold representing a %-age of the distance from a perfect match) and "ambiguous match ratio" (used to reject ambiguous matches) were 5 m, 100 and 0.6, respectively. Once again, the parameter values were obtained empirically.

***Geographical Transformation and Mosaicking:*** A 2D geometric projective transformation [62] is used to convert each swath, $S^i$, to its correct location and orientation, $T^i$. The process requires at least four matching feature pairs so that the resultant transformation model, $t^i$, can account for translation, scale, shear and rotation. However, unlike the affine transformation, where parallel lines are constrained to remain parallel, for a projective transformation, parallel lines are permitted to converge to a vanishing point. Generally, the more feature matches correctly identified, the more accurate the transformation.

By applying $t^i$ to each spectral band, RGB images, hyperspectral indices and entire data cubes may be created through the progressive placement of georectified swaths into the (initially blank) mosaic composite matrix, with one mosaic per band stacked as a raster cube. Pixels of the cube or RGB mosaic that overlap with one another may be determined by using the value from the last swath added to the mosaic or by applying a mask to the intersection of the current swath and existing mosaic and averaging the relevant pixels before combining them to the mosaic. The latter is preferred.

***Creation of 3D Dense Point Cloud:*** Following the rectification and alignment of the hyperspectral images to the RGB orthomosaic—which was derived from the DPC (in

MetaShape) and is a matrix of dimension $[p, q, r]$—a grid-averaged raster is formed. The process computes an axis-aligned bounding box for the entire point cloud, dividing it into $p \times q$ grid boxes of size GRD. Points within each grid box are merged by averaging their locations, colours and surface normals. As the orthomosaic and rastered DPC are already aligned, it is then a simple process to align the hyperspectral mosaic to the grid averaged DPC by allocating height and surface normal properties to each pixel. This provides a dense hyperspectral point cloud.

## 5. Experimental Results

The accuracy of the results was evaluated using a combination of qualitative and quantitative factors. The qualitative assessment of the georectified hyperspectral mosaics was carried out by visually inspecting the imagery for gaps, mismatches, deformations, duplicates and discontinuities (Figure 6). Initially, no manual adjustment was applied, i.e., Figures 6–16 show results for the automated workflow only.

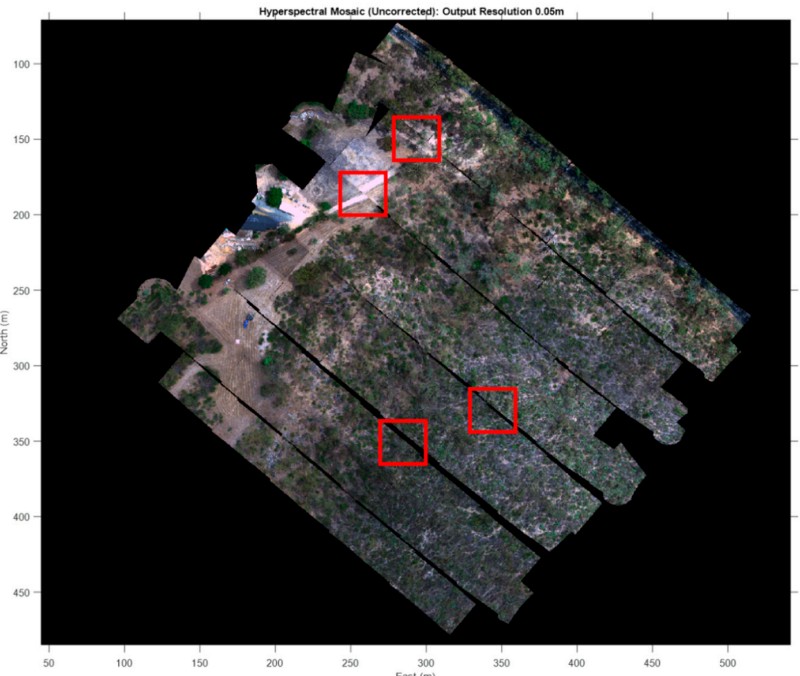

**Figure 6.** Uncorrected hyperspectral mosaic formed from raw GNSS data. The red boxes indicate regions from which the zoomed regions are taken and shown in Figures 7 and 8.

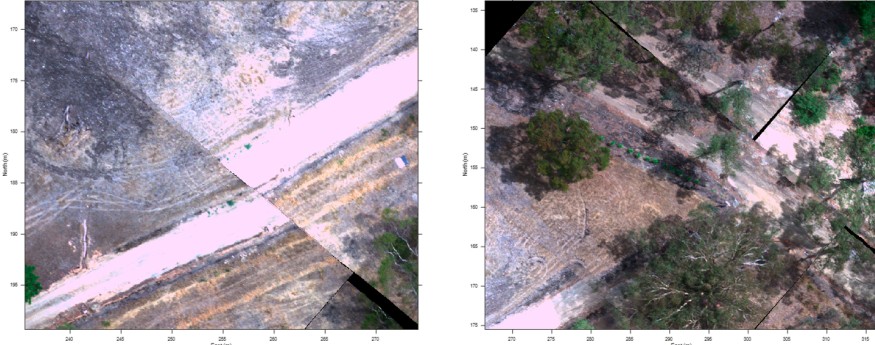

**Figure 7.** Zoomed regions of Figure 6, showing (**left**) mismatched features between swaths and (**right**) features observed twice in contiguous images.

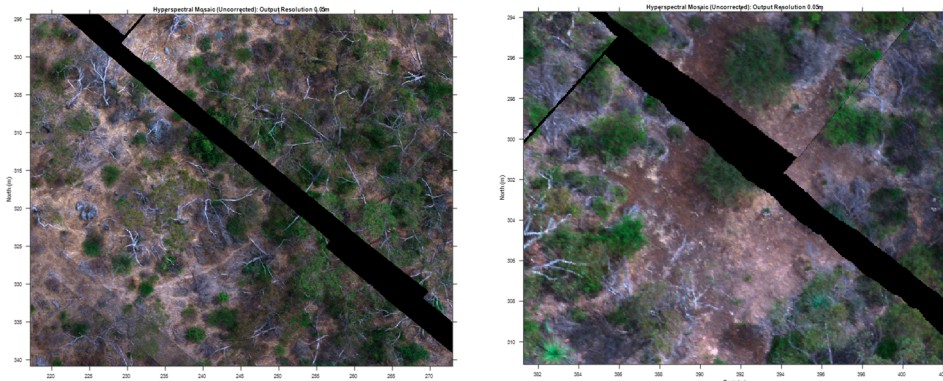

**Figure 8.** Zoomed regions of Figure 6, showing gaps between contiguous along- and across-track images.

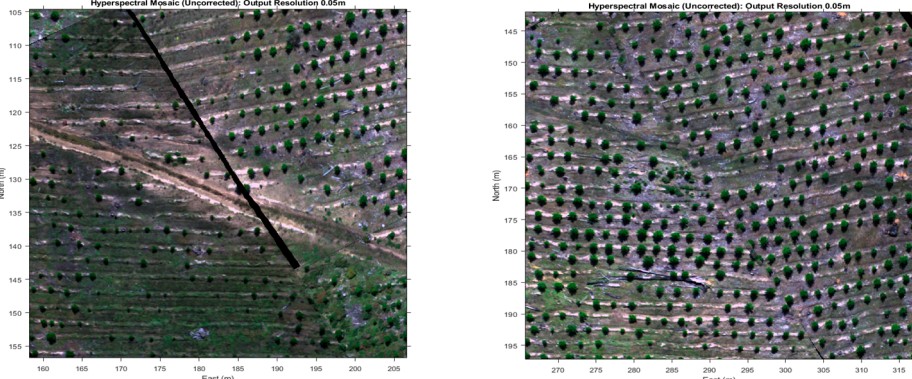

**Figure 9.** Zoomed regions of the Goat Farm B site showing discontinuities in lines of two-year-old trees.

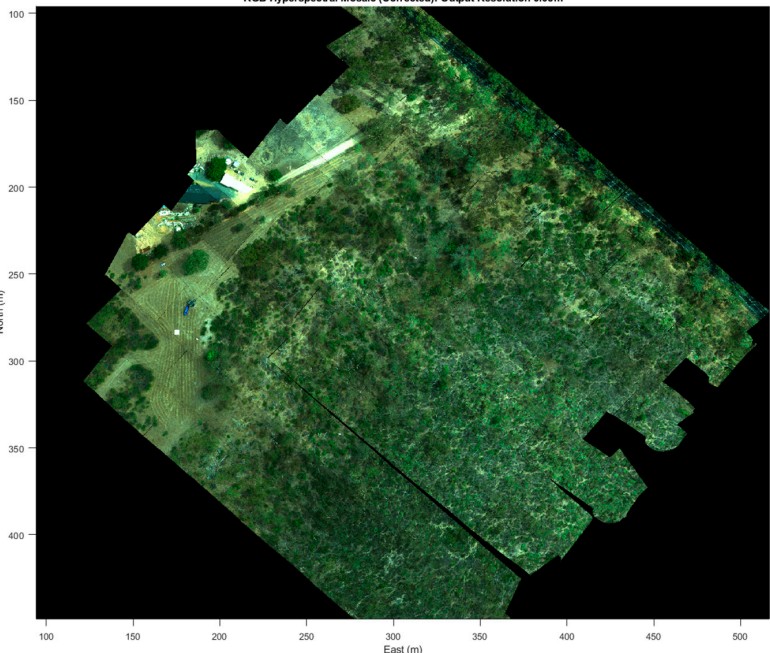

**Figure 10.** Georectified hyperspectral mosaic formed using the workflow described in this paper (note: a few single images at the ends of some swaths shown in Figure 9 have been auto rejected). Also, the very narrow side overlap (<5%) has resulted in a few gaps between swaths.

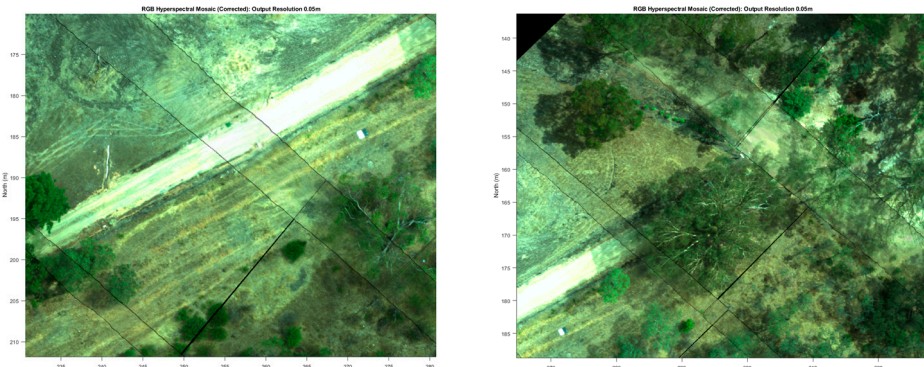

**Figure 11.** Zoomed regions of the georectified hyperspectral mosaic (taken from Figure 10), showing the improved alignment of features between swaths and features that were observed twice.

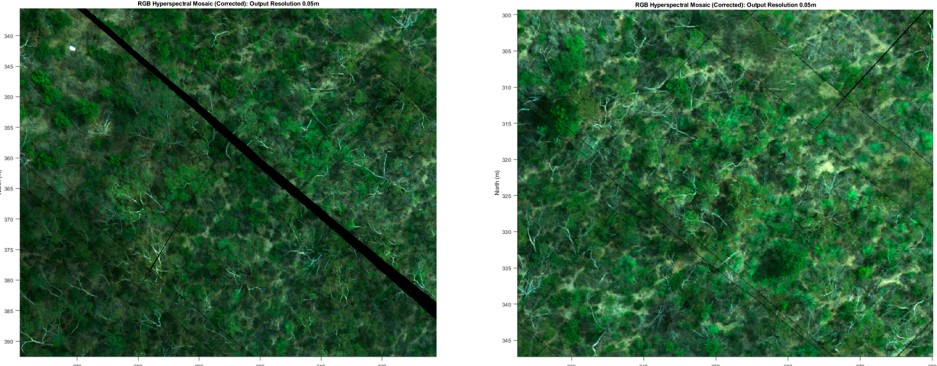

**Figure 12.** Zoomed regions of the georectified hyperspectral mosaic (taken from Figure 10), showing the improved edge alignment for contiguous images in both the along- and across-track directions.

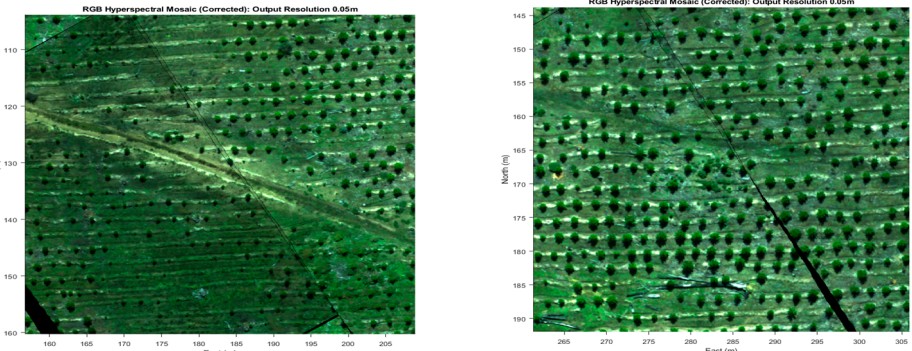

**Figure 13.** Zoomed regions of the Goat Farm B data set showing georectified lines of two-year-old trees. These images should be compared to those shown in Figure 9.

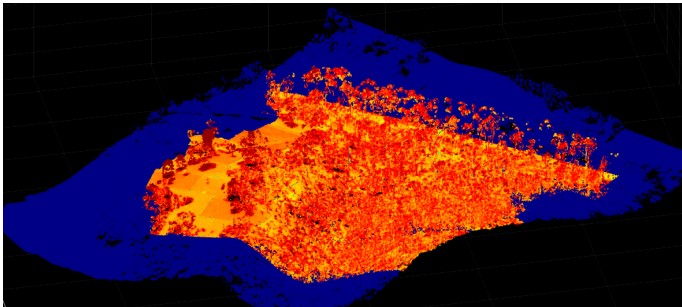

**Figure 14.** The 3D point cloud of NDVI for the PEGS Hale site. The elevated levels of NDVI can be seen to accurately align with the vegetation structures such as the tall trees (at the back of the site).

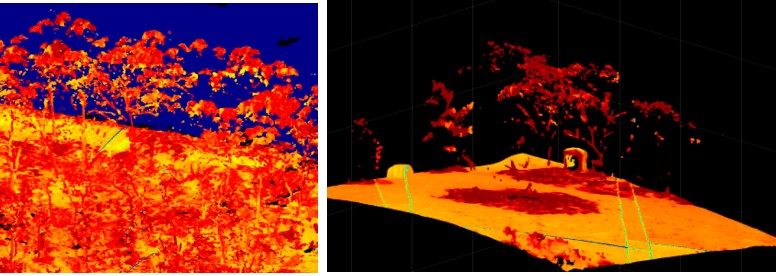

**Figure 15.** Zoomed regions of Figure 14 showing the 3D point cloud of NDVI for the PEGS Hale site.

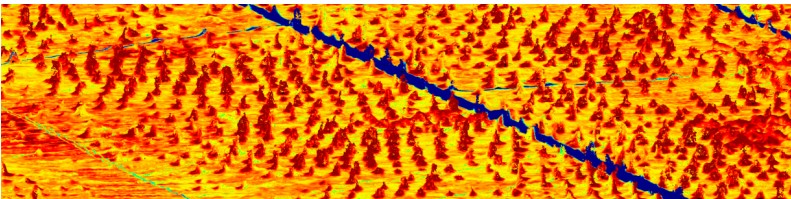

**Figure 16.** Zoomed regions of the Goat Farm B site showing the NDVI point cloud derived from the auto-rectified workflow process. The darker regions depict higher values of NDVI, which align well with the 3D structures indicating young trees. The lighter regions depict lower values of NDVI, which generally correspond with the surface of the terrain.

Figure 7 (left) shows a mismatch across an image boundary. Figure 7 (right) shows duplicate features in neighbouring across-track images. Figure 8 (left and right) shows gaps and discontinuities in contiguous image sequences. Figure 9 (left and right) shows discontinuities in the lines of young trees. Comparing these to the georectified mosaic derived from the workflow described in this study (Figure 10), and particularly the zoomed regions (Figures 11–13), shows that linear features such as roads, gullies and tree lines now have continuity across and within swaths, and the visual consistency, sizes and shapes of objects such as trees and branches are also well-maintained (note: the text in these figures is best viewed in a 'zoomed' version of the manuscript online). As mentioned in the section on hyperspectral data acquisition, however, the low swath overlap combined with sensor misalignment errors, still results in some gaps between contiguous swaths (note: the faint lines between individual images in Figures 12 and 13 indicate the edge of each image. They can be removed using interpolation).

The qualitative assessment of 3D DPC was carried out by visually inspecting point clouds colour coded using the normalized difference vegetation index (NDVI) and enhanced vegetation index (EVI). Accurate georectification and mosaicking of the hyperspectral images will result in high levels of NDVI/EVI (vegetation) coinciding with higher z-axis (height) values of the DPC. In Figure 14 (and Figure 15), it can be seen that this is the case: the tall trees at the rear of the site and the shrubs and flora scattered around the site more generally all have high values of NDVI (depicted by darker red colours in Figures 14 and 15), whereas the lower levels of NDVI (the orange-coloured elements) correspond to the surface of the terrain.

Figure 16 shows the NDVI 3D DPC for the site at Goat Farm B, which contains repetitive scene structures of young trees. The trees, which form a regular repetitive pattern over the ground (and have high values of NDVI), closely correspond to the small (elevated) bumps in the DPC, which also correspond to the trees.

Adjusting the auto-rectified hyperspectral mosaic using the manually identified reference points (as described above) made no discernible improvement to the qualitative assessment of either the 2D mosaic or 3D DPC, except that there was a slightly better alignment between the higher NDVI values and elevated structures in the DPC. In other words, there were fewer poorly aligned tree structures slightly offset from their higher

NDVI values (as can be seen for the auto-rectification approach in the upper righthand corner of Figure 16).

Quantitative assessment of the relative positional accuracy of the auto-georectified hyperspectral mosaics is the most important factor for evaluating the workflow. The Euclidean distance between the east and north coordinates of randomly selected pairs of reference points, identified across both the georectified RGB hyperspectral and original orthomosaic images, were compared. Basic statistical metrics (mean absolute error (MAE), RMSE and circular error probable (CEP) at the 95% confidence level) were then used to characterise the relevant error distributions. Using 50 randomly distributed reference points, the accuracy achieved for each of the sites is shown below in Table 1.

**Table 1.** Auto-georectification accuracy and processing times by site (mean absolute error, root mean squared error and circular error probable at the level of 95% confidence).

| Site | Hyperspectral Mosaic Dimension (pixels) | Global Accuracy (m) | | | Net Processing Time Mosaic/DPC (mins) |
|---|---|---|---|---|---|
| | | MAE | RMSE | CEP$_{95\%}$ | |
| Goat Farm A | 99,551 × 9550 | 0.55 | 0.36 | 0.65 | 14 (+3 for DPC) |
| Goat Farm B | 11,100 × 10,354 | 0.49 | 0.27 | 0.78 | 15 (+3 for DPC) |
| Hale (PEGS) | 10,443 × 12,510 | 0.87 | 0.56 | 0.87 | 18 (+6 for DPC) |
| Bonython (PEGS) | 8096 × 9729 | 0.89 | 0.51 | 0.84 | 13 (+3 for DPC) |
| **Mean Values** | **10,365 × 10,805** | **0.70** | **0.42** | **0.78** | **15.0 (+3.7 for DPC)** |

It is worth noting that any tool that can be used to assess discrepancies between images, mosaics or DPCs can also be used to improve alignment between them. Consequently, prior to the creation of the 3D DPC and assessment of any automated georectification method, the software used to instantiate the workflow described in this study offers users the opportunity to manually identify matching reference points between the hyperspectral and RGB orthomosaic images. As the tool (provided only in MATLAB) offers predictive point projections, and the images are already accurately mosaicked, the activity only takes about 2 min per 10 pairs of reference points.

These manually identified control points may then be used to reduce any residual discrepancies between the two mosaics. Piecewise-linear transforms are used, such that the hyperspectral mosaic is broken into local piecewise-linear regions and a different affine transformation applied to each region. The accuracy achieved after this manual process is shown below in Table 2.

**Table 2.** Resultant manual georectification accuracy by site (after auto-alignment). The results are based on the identification of 50 reference points arbitrarily (but uniformly) distributed across each site. The identification of the 50 pairs of reference points took about 10 min.

| Site | Hyperspectral Mosaic Dimension (pixels) | Global Accuracy (m) | | |
|---|---|---|---|---|
| | | MAE | RMSE | CEP$_{95\%}$ |
| Goat Farm A | 99,551 × 9550 | 0.33 | 0.25 | 0.57 |
| Hale (PEGS) | 10,443 × 12,510 | 0.67 | 0.41 | 0.61 |

***Computational Efficiency:*** A key criterion of any automated technique is its computational efficiency, which is usually expressed in terms of how long it takes to process data on a particular machine. The workflow should be faster, more convenient and more robust than the alternatives. Unfortunately, very few methods report their computational efficiency, with Angel et al. [8] as an exception. We therefore compare our approach to theirs.

The workflow was entirely coded in MATLAB and executed on a Dell laptop with 32 GB RAM and an Intel® Core i7-8850H CPU (6 cores) running at 2.59 GHz. Angel et al. [8] ran parallelised MATLAB code on a desktop with an Intel Xeon-2860 v2 processor that had 200 GB RAM and 20 cores running at 2.8 GHz: a considerably more powerful machine. Unlike [8], code parallelisation was not used in this study and the laptop had only a modest GPU (Intel® UHD Graphics 360).

The amount of user and computational effort required to manipulate data is highly dependent upon the size of the site and GRD. In particular, computational effort is closely related to site area/GRD. Both studies created mosaics of roughly the same size, around $10^4 \times 10^4$ pixels. For mosaics of this size, manual co-registration requires about 3 min of user effort per pair of identified reference points. Thus, for a typical site used in this study (90 individual hyperspectral images), if eight pairs of reference points are identified per image, the manual georectification task will take around 36 hours' continuous effort, i.e., almost a standard working week of full-time effort (note: although we did not perform formalised time testing, we did manually align the Goat Farm data sets and this amount of projected effort derives from our experience). Alternatively, if effort is expended upfront generating individual swaths during the pre-processing stage, and 27 matching pairs of points are needed to align each of these to the orthomosaic (as per [8]), around 20 h of user effort is required if the 90 images form 15 swaths.

Using the workflow devised by [8], which starts with the creation of individual hyperspectral swaths, this is reduced to 3.7–6.5 h. However, it is unclear how much additional effort they required to create the individual hyperspectral swaths during the pre-processing stage. If the SpectrononPro3 software had been used, considerable additional user effort would have been needed: the hyperspectral data sets must be broken into contiguous sequences of raw data commensurate with each swath (groups of consecutive files); and yet more effort is needed if pitch, roll, yaw and timing misalignment offsets need to be optimised before any swath pre-processing takes place.

To increase the degree of automation, we created a "batch processing" approach that requires very little user effort. The pre-processing stage can simply deliver its output to a single folder and the workflow then processes each file/cube individually, accurately geo-rectifying and aligning the output into the mosaic and DPC. Overall, the task of importing all the data, georectifying and aligning it into a hyperspectral mosaic and then displaying them as output is reduced to around 15 min, with the generation of hyperspectral 3D DPC only taking a further 3–6 min (Table 1). This represents a significant improvement in computational efficiency over previous automated and semi-automated techniques.

## 6. Discussion

There are several different approaches to georectifying, aligning and mosaicking UAV-based hyperspectral pushbroom data. Those making use of co-registration techniques with RGB-based orthomosaics tend to perform better in terms of accuracy and output than those reliant upon dense networks of GCP [21,23]. The most accurate automated approach we are aware of is that of Angel et al. [8], who achieved MAE and RMSE values of around 7 and 10 times their GRD, respectively (but as good as 1–1.5 times). We achieved average ratios for MAE and RMSE of around 13 and 8, respectively.

As the RGB orthomosaic and hyperspectral mosaics are co-registered to the same pixel framework, a visual assessment of georectification and alignment results is readily achieved by repeatedly switching between images superimposed over one another. Undertaking qualitative examination of the results in this way indicates that the error distributions are generally evenly distributed (as per [8]) across the sites. However, this visual examination also indicates that there are regions of very high accuracy contrasted with regions that have a lower accuracy. This regional accuracy/inaccuracy is also borne out by visual examination of the structure of the DPC and hyperspectral mosaics.

The prime significance of the above is that, whilst global accuracies for the technique suggest that mosaics and DPCs may not align sufficiently well to assist in modelling complex surface reflectance characteristics under real-world conditions, regions of the DPC and hyperspectral cubes clearly are accurately aligned. Furthermore, these regions are readily identified by visually toggling between the reconstructed and target mosaics.

The regional accuracy does not appear to be related to the number of matched features, but more the constraints that result from a projective transform applied to an entire swath. This suggests that accuracy gains may be available by reducing the number of images processed

together (as swaths) prior to application of the feature detection and matching stages. It also suggests that targeted manual co-registration using the MATLAB tool CPSELECT (referred to above) could offer regional improvements. This is the focus of ongoing research.

In theory, SURF and MSER are agnostic to textural variations caused by luminance differences between images. In practice, however, similarity in luminance values across data sets provides an improved performance during the feature matching stage. Although [8], like this study (but unlike earlier ones), establishes a comparison between hyperspectral and RGB imagery data (earlier studies often used only a single band), white balance or luminance values are not normalised or rescaled. This additional manipulation improves the efficiency and performance of SURF/MSER and MLESAC during the feature detection, co-registration and matching stages. Typically, between 1000 and 4000 matches are generated per swath, i.e., around 15,000–50,000 matches per mosaic. Such an approach is vital to the creation of an accurate projective transform model.

Additionally, it also accounts for (some) variation in the lighting that can occur between the hyperspectral and RGB sensor flights, as well as radiometric differences. It thus offers additional operational flexibility. That is, whilst it is ideal for flight speed and altitude, frame rate sampling and FOV to enable data to be observed at a roughly equal spatial resolution to the RGB sensor, using the workflow described here reduces the need for simultaneous (or near-simultaneous) data collection, i.e., the need for data capture under roughly similar atmospheric, and thus illumination, conditions is relaxed. Similarly, although the GRD of the RGB and hyperspectral sensors almost always differ a little in practice, because resampling is an intrinsic component of our workflow (to align the mosaic pixel indices), features appear in each image set at roughly comparable scales. These factors combine to simplify flight planning.

Other researchers have shown that using only a few GCPs, high precision base stations, and good GNSS-IMU sensors integrated with the cameras are needed to produce high-quality results. For instance, [63] showed that 3 GCPs/Ha were sufficient to deliver sub-centimetre accuracies, and [22] likewise achieved similar accuracies with only a few GCPs.

Although at least six AeroPropeller auto-GCPs (www.aeropropeller.com) were deployed at each site to accurately georectify the orthomosaic and DPC, as the absolute position of the MetaShape products within the WGS84 frame of reference was unimportant to this study, the GCPs were not used in the workflow. In other words, whilst the GCP could have been manually or auto-identified by MetaShape, and the RGB orthomosaic then more accurately composed relative to an absolute framework, this was not undertaken because any georegistration improvement in the orthomosaic would not have contributed to the outcome of this study. Moreover, and perhaps more importantly, visual analysis of the SURF and MSER feature detection and matching showed that the use of the GCPs did not influence the performance of the method in terms of its automation or accuracy (there were typically in excess of 1000 good feature matches per swath between the two image sets). Also, the use of GCPs that auto-register within a first-order survey network impacts only on the time taken to generate the RGB orthomosaic using MetaShape and not our workflow.

Another facet of the data sets used in this study is that, whilst the Goat Farm data comprises young trees, which like the tomato crops and date plantations used in other studies are planted in regular lattice-like structures, the features of the PEGS data sets are more irregular and ill-structured. They are therefore more susceptible to textural variability between the RGB and hyperspectral imagery. Such effects result from the variability of (for instance) shadows cast by trees. In the PEGS data sets, which comprise naturally occurring flora as opposed to strategically planted young trees, the shadows are typically cast in a less well-formed manner, both in terms of contrast and shape. This typically impacts the feature detection and matching stages negatively, and this appears to be borne out by the accuracy results (Table 1).

In terms of computational efficiency, the focused application of SURF and MSER to the (masked) elements of the RGB orthomosaic that are most likely to contain matching features significantly reduces the image processing load relative to earlier approaches [6,8].

Furthermore, based on descriptions by researchers who relied on PARGE [43,64] to deliver sub-metric accuracies (4 h for georectified airborne scans comprising 200 spectral lines for $10^6$ pixel mosaics), it is thought that the workflow described here requires considerably less computational and user effort. In particular, significantly less user effort is thought to be required during pre-processing as our workflow intentionally employs a batch processing approach, with georectification and mosaicking taking only 18–24 min, depending on whether or not a 3D DPC is also required.

Angel et al. [8] made use of MetaShape's *ultra-high* levels of DPC resolution. In our experience, whilst the process is largely automated, these very-high-quality DPCs take a very long time to synthesise (often in excess of 24 h), even on high-performance computers. The study reported here used only *high* resolution DPCs, which typically take only around 2 h to generate for sites comprising 100 images.

Similarly, executing SpectrononPro3 in its batch processing mode takes around 40 min to process 100 hyperspectral data cubes and place the output into a folder, ready for input to the workflow reported in this manuscript. Considerable additional user effort would be needed to break contiguous data sequences into file sets commensurate with each swath, i.e., in preparation for the approach used by [8]. For instance, for a data set of 100 pushbroom hyperspectral cubes forming 15 swaths and with data extraneous to the region of interest, it is estimated that such a task would take of the order of 2–3 h. This suggests further potential time savings are available relative to the approach adopted by [8].

Finally, even though field data indicates that we describe an automated technique for georectifying, aligning and mosaicking hyperspectral pushbroom data that is simultaneously computationally efficient and accurate, there are a few improvements that could enhance its performance. The first is code parallelisation. Despite the workflow not employing parallelised code, this could easily be attempted. However, it should be noted that the mosaicked imagery is progressively built from composite images using matrix or pixel index operations. Parallelisation would thus likely require a computer with considerably more RAM.

Further work could also examine the use of GCP networks, and in particular the auto-detection of the individual boards, over several study sites to examine the absolute position accuracy of hyperspectral mosaics. There may also be merit in comparing a variety of feature identification techniques (SIFT, SURF, BRISK, ORB, KAZE, etc.) to see which deliver optimal performance. However, the results would likely be influenced by the test data set, resulting in the need for it to be significantly expanded. Furthermore, there may also be merit in arriving at parameter sets for the SURF, MSER and MLESAC algorithms used in this study by more methodical means than the empirical methods used herein. A comprehensive cross-comparison of parameters (in particular against varying data sets) would potentially deliver performance enhancements. Once again, however, such an undertaking would be significant.

## 7. Conclusions

A processing pipeline for swiftly and automatically generating, fusing, georectifying and rendering dense hyperspectral point clouds is presented. The proposed methodology uses images acquired using a pushbroom hyperspectral sensor carried by a UAV and assembles them into georectified 2D mosaics before displaying them as a 3D DPC. The technique relies upon an auxiliary RGB orthomosaic collected by a UAV carrying a frame-based camera.

The technique substantially reduces both the raw pre-processing requirements and computational load of workflows that existed previously, thereby increasing the potential operational utility of the method. To reduce the pre-processing requirements, the proposed method morphologically and geometrically assesses (and, if possible, repairs) hyperspectral data cubes before aligning them into linear swaths. To reduce the processing load in the image processing component of the workflow, luminance normalisation and the targeted application of vision processing strategies are used to improve the performance of the SURF, MSER and MLESAC feature detection and matching algorithms.

The result is a workflow that is simultaneously computationally efficient, robust, easy-to-use, delivers high spatial accuracy and enables hyperspectral analysis in 3D space. Hyperspectral mosaics with a 5 cm spatial resolution were mosaicked with relative positional accuracies characterised by MAE, RMSE and $CEP_{95\%}$ of 0.70 m, 0.42 m and 0.78 m, respectively. The technique was tested on five scenes comprising two types of landscape.

There are opportunities for improving the computational efficiency of the technique by using code parallelisation. However, these methods are currently thought to require computers with more memory, which might reduce the general accessibility of the method.

**Author Contributions:** Conceptualization, A.F.; methodology, A.F., S.P. and P.K.; software, A.F.; validation, A.F., S.P., P.K. and J.O.; formal analysis, A.F., S.P. and P.K.; investigation, A.F., S.P., P.K. and J.O.; resources, J.O.; data curation, P.K. and A.F.; writing—original draft preparation, A.F.; writing—review and editing, A.F., S.P., P.K. and J.O.; visualization, A.F., S.P., P.K. and J.O.; supervision, A.F.; project administration, J.O.; funding acquisition, A.F., S.P. and J.O. All authors have read and agreed to the published version of the manuscript.

**Funding:** Funding for this study was made available by NIFPI Project NS020, "Solutions for the optimal use of remotely acquired high resolution data by the forestry sector".

**Data Availability Statement:** An executable version of the software used to generate the results published in this paper is available without charge. Contact anthony.finn@unisa.edu.au for details.

**Acknowledgments:** The authors are very grateful to the Mount Gambier Centre of the National Institute for Forest Products Innovation (NIFPI) for their continuing support and assistance. We are also grateful to Steve Andriolo of EyeSky, who conducted the Mount Crawford and PEGS UAV operations and Paul Markou of the Mount Crawford branch of ForestrySA for providing access to the Goat Farm site and his assistance throughout the field trials. We are also deeply grateful to Arko Lucieer and Darren Turner of the University of Tasmania for their preliminary feedback.

**Conflicts of Interest:** The authors declare that they have no known competing financial interests or personal relationships that could have appeared to influence the work reported in this paper.

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
