# Peer review of "Automated Georectification, Mosaicking and 3D Point Cloud Generation Using UAV-Based Hyperspectral Imagery Observed by Line Scanner Imaging Sensors"

_remotesensing, doi:10.3390/rs15184624_

Round 1

Reviewer 1 Report

In this article, it is proposed to  a fast, automated and computationally robust georectification and mosaicking technique that generates 3D hyperspectral point clouds. Generating point clouds from photographs is a hot topic, generating point clouds from hyperspectral images and the mosaicing process is the focus of this article. Thanks for your good idea.

1- 'Section 3.4 of [8] for surf', 'Section 3.5 of [8] for mlesac'. These sections are referred to in the article, but their mathematics is not mentioned. The math of these algorithms should be added.

2-      Additions to the literature should be made.

3-      28 29 30 articles not cited.

4-      Mathematical models of methods should be given as formula, with formula numbers. It is very difficult to understand this way.

5-     The results should be given as tables, you should give the mathematical difference with the results of other methods ( For example: shift, Brisk etc. Algorithms) in the form of tables. The results should be compared with other algorithms or compared with normal images. if you claim that the method is a new method, you should clearly show it with tables. For example, How many million point loud data was generated?

6-      Why you choose surf and MSER?

Author Response

Please see detailed responses in file

Reviewer 2 Report

No comments

Minor editing of English language required

Author Response

Please see detailed reposnses in file

Reviewer 3 Report

The authors describe a novel fast, automated and computationally robust georectification and mosaicking technique that generates point clouds using hyperspectral images observed from UAVs with 3D hyperspectral Push-Broom sensors. The authors' intention is clear but the description is rather confusing in different parts of the paper. Consequently, I suggest a revision of the paper in order to make it easier to read and to understand the improvements made in the field with respect to what is found in the literature. Below, the authors can find some suggestions and comments:

TITLE

I suggest revising the title since the name "push-broom" is not current. Therefore, I suggest replacing this name in line scanner imaging systems. I suggest viewing and adding in the paper the reference: https://doi.org/10.1080/22797254.2018.1444945

ABSTRACT

I suggest shortening the extension of the abstract by emphasising the research results.

INTRODUCTION

Line 49                I suggest to add a reference.

RELATED WORK

The acronyms GNSS and IMU must be specified

Line 91-93          The sentence is questionable, so I suggest revision or deletion

Line 102              I suggest adding the following reference: https://doi.org/10.5194/isprs-archives-XLIII-B1-2022-325-2022

METHOD

In the method description, there are subsections with equations and numbers that seem to be incomprehensible. Therefore, I suggest adding some figures or better specifying the variables describing the equations. Is it strictly necessary to include figures 4 and 5? In my opinion, these figures do not add quality to the paper. The equations must be described according the guidelines of the journal.

EXPERIMENTAL RESULTS

Pag.13 I suggest reviewing the numbering of the pictures because there is something strange; please check. Also, I suggest describing figure 6 first and then inserting the image.

The text of pictures 10 to 12 is not readable.

figure 14 is not mentioned in the text and a scale of values needs to be introduced (or at least a description of the legend added)

REFERENCE

It is necessary to format the paper according to the guidelines of the journal

Author Response

Please see detailed responses in file

Round 2

Reviewer 1 Report

The authors have not contributed to the algorithms of Surf, MSer and Mlesac. Yes, I can understand that. However, the authors used these algorithms in Matlab and performed the point cloud and mosaicing process according to these algorithms. Therefore, the mathematics of these algorithms should be briefly explained.

More references have been added, and a few desired corrections have been made. However, the author's statement, "the aim of this research is to minimize human intervention during the geoprocessing stage, avoid reliance on GCP, and fully automated an efficient 'batch processing' co-registration, mosaicking and point cloud generation strategy. It implements a simplified co-registration strategy." co-registration and mosaicking strategy and generates positionally accurate 3D hyperspectral point clouds; and it does so in a computationally efficient manner." This explains the purpose of the study. SURF, MSER and MLESAC are algorithms that affect the accuracy of this study. Therefore, explaining these is not unnecessary; on the contrary, it increases the understandability of the article. In my opinion, this is the most critical issue to understand after the mosaicing process. Threshold values in the Surf, MSer or MLESAC algorithms will affect your work's global accuracy and transformation accuracy values.

I would like to ask you to reconsider the results and narrative of your study according to these values. I think the work needs a little more improvement, and for this reason, I reject it.

Reviewer 3 Report

I am glad that the authors responded to all my comments and suggestions. The paper can be accepted in this form. 

Author Response

We thank the reviewer for their helpful comments